# A Hierarchical Framework for Assessing Transmission Causality of Respiratory Viruses

**DOI:** 10.3390/v14081605

**Published:** 2022-07-22

**Authors:** Tom Jefferson, Carl J. Heneghan, Elizabeth Spencer, Jon Brassey, Annette Plüddemann, Igho Onakpoya, David Evans, John Conly

**Affiliations:** 1Department for Continuing Education, University of Oxford, Rewley House, 1 Wellington Square, Oxford OX1 2JA, UK; igho.onakpoya@conted.ox.ac.uk; 2Nuffield Department of Primary Care Health Sciences, University of Oxford, Radcliffe Observatory Quarter, Woodstock Road, Oxford OX2 6GG, UK; carlheneghan@phc.ox.ac.uk (C.J.H.); elizabeth.spencer@phc.ox.ac.uk (E.S.); annette.pluddemann@phc.ox.ac.uk (A.P.); 3Trip Database Ltd., Little Maristowe, Glasllwch Lane, Newport NP20 3PS, UK; jon.brassey@tripdatabase.com; 4Li Ka Shing Institute of Virology, Department of Medical Microbiology & Immunology, University of Alberta, Edmonton, AB T6G 2R3, Canada; devans@ualberta.ca; 5Centre for Antimicrobial Resistance, Alberta Health Services, Alberta Precision Laboratories, University of Calgary, Calgary, AB T2N 4N1, Canada; john.conly@albertahealthservices.ca

**Keywords:** viral transmission, causation, evidence hierarchy, SARS-CoV-2, respiratory pathogens

## Abstract

Systematic reviews of 591 primary studies of the modes of transmission for SARS-CoV-2 show significant methodological shortcomings and heterogeneity in the design, conduct, testing, and reporting of SARS-CoV-2 transmission. While this is partly understandable at the outset of a pandemic, evidence rules of proof for assessing the transmission of this virus are needed for present and future pandemics of viral respiratory pathogens. We review the history of causality assessment related to microbial etiologies with a focus on respiratory viruses and suggest a hierarchy of evidence to integrate clinical, epidemiologic, molecular, and laboratory perspectives on transmission. The hierarchy, if applied to future studies, should narrow the uncertainty over the twin concepts of causality and transmission of human respiratory viruses. We attempt to address the translational gap between the current research evidence and the assessment of causality in the transmission of respiratory viruses with a focus on SARS-CoV-2. Experimentation, consistency, and independent replication of research alongside our proposed framework provide a chain of evidence that can reduce the uncertainty over the transmission of respiratory viruses and increase the level of confidence in specific modes of transmission, informing the measures that should be undertaken to prevent transmission.

## 1. Introduction

“*Our increasing proficiency in demonstrating viruses has produced a disconcerting but not entirely unwelcome paradox—the spectacle of new information leading to confusion.*” [1]

The COVID-19 pandemic has necessitated widespread global action, and research to inform evidence-based policy decisions has been urgently needed. Understanding the circumstances of SARS-CoV-2 transmission has been a public health priority. Yet, a consensus is lacking, partly due to the suboptimal quality of the available research to date [2,3,4,5,6,7,8]. The lack of high-quality evidence has highlighted the pressing need for a contemporary conceptual framework to assess causality in the transmission of respiratory viruses in humans, especially SARS-CoV-2. 

The background to this work initially was a series of systematic reviews funded by the WHO on modes of transmission of SARS-CoV-2, including airborne, fomite, orofecal, and the association of close contact with transmission [9,10,11,12,13]. The reviews revealed significant methodological shortcomings in the included studies with a lack of standard methods in the design, conduct, testing, and reporting of SARS-CoV-2 transmission and a consistent lack of proof of replicability of the results.

Here we do not argue for or against any particular mode of transmission. Rather, we briefly review the history of causality assessment related to microbial etiologies with a focus on viral etiologies and then suggest a hierarchy of evidence to integrate clinical, epidemiologic, molecular, and laboratory perspectives on transmission. The hierarchy, if applied to future studies, should narrow the uncertainty over the twin concepts of causality and transmission of human respiratory viruses. These distinct concepts have become intertwined by the advent of genomics, as our historical and evidence narrative will show.

## 2. From Bacteriology to Virology 

The first formulation of principles of causality between a microbe and disease is that by the German physicians and microbiologists Robert Koch and Friedrich Löffler [14] (Table 1). Although dated, the postulates are an essential first attempt at setting out attribution rules and are the definitive move away from miasma theory previously prevalent [15]. 

The next significant contribution was by the American bacteriologist and virologist Thomas Rivers, who, in 1936, analysed the implications on the postulates in the emergent fields of virology and immunology [16]. Rivers formulated two simple postulates: (1) a specific virus must be found associated with a disease with a degree of regularity; and (2) the virus must be shown to occur in the unwell individual not as an incidental or accidental finding but as the cause of the disease under investigation. Rivers addressed the historical shortcomings of Koch’s work—and their subsequent distortions—by recognising the viral agent may not necessarily be present each time a case of the particular “malady” was observed. The originality of Rivers’ contribution lay in separating causality from association and recognising the possibility of contamination or co-infection (a point he makes in an elegant discussion of the etiology of encephalitis lethargica or Von Economo’s disease, and the coincidental finding of herpes virus in some of the slides from those cases) [16].

As virology developed further, the issue of differentiating between causation and association led the American physician and virologist Robert Huebner, in 1957, to issue a “Bill of Rights for Prevalent Viruses, a guarantee against the imputation of guilt by simple association” [1]. Huebner pointed out the paradox of the advancement of virological knowledge exemplified by noting, “our increasing proficiency in demonstrating viruses has produced a disconcerting but not entirely unwelcome paradox—the spectacle of new information leading to confusion” [1]. 

Huebner asked the fundamental question of how can we tell what is causing what? He proposed eight points to assess viral causality (Table 2):

The development of virological techniques is reflected in the detailed and multidisciplinary approach to the exclusion of a simple association. Each of his points, per se, is insufficient to prove or disprove anything. However, if taken in their entirety, they would narrow the chance of an association and strengthen the case for causality.

In 1973, the Yale epidemiologist Alfred Evans reformulated Rivers’ postulates based on up-to-date immunology theory and the discovery of antigens and listed seven immunologic criteria related to proof of causality which complemented Rivers’ postulates [17]. Meanwhile, the statistician Austin Bradford Hill in 1965 had introduced what he called criteria for causality in another President’s address [18].

## 3. Hill’s Criteria

Hill, a British statistician by training—famous for his work on tobacco and lung cancer—proposed nine criteria to assess the nature of the association between two variables: strength, consistency, specificity, temporality, biological gradient, plausibility, coherence, experiment, and analogy [18]. Hill stated there was no magic formula to assess causality and his criteria were a guide or “viewpoints” and did not provide “indisputable evidence for or against the cause-and-effect hypothesis”. A degree of subjectivity was necessary in interpreting the evidence and no viewpoint was a “*sine qua non*”.

## 4. Gwaltney and Hendley Postulates

In 1978 Jack Gwaltney and J. Owen Hendley, based at the University of Virginia, proposed postulates based on their work with rhinovirus and other respiratory viruses to address respiratory virus causality (Table 3) [19]. 

The first four postulates can be considered a practical application of Hill’s criteria of strength and temporality, but the fifth is Hill’s experiment: “Because of an observed association some preventive action is taken. Does it, in fact, prevent? Is the frequency of the associated events affected?” [19]. 

The postulates present the first convergence of causality and transmission as they represent a detailed checklist of the chain of transmission of respiratory viruses and require an experiment to test whether the hypothesised mode of transmission can be interrupted. 

The Gwaltney–Hendley postulates, however, omit Hill’s criteria of “specificity” and “consistency”. Specificity is a particular problem concerning viral respiratory infections. Since the clinical presentation cannot distinguish those viral infections that present with similar clinical features, laboratory identification of the responsible agent becomes necessary. Evans pointed this out in 1991 in his “Five Realities of Acute Respiratory Disease” (Table 4) [17]. 

As any practising clinician knows, it is impossible to distinguish with absolute confidence an acute respiratory infection caused by rhinovirus from one caused by adenovirus, or to be able to rule out any co-infection with two or more viruses, based on clinical presentation alone. Proof requires microbiological identification and evidence for the chain of transmission, which is now possible with the advent of genomics [20].

## 5. The Genomics Era 

As knowledge of genetics grew, Fredricks and Relman in 1996 proposed reformulation of the original Koch’s postulates incorporating the state of knowledge at the time (Table 5) [21]. 

This list contains many previously raised points, and, crucially, includes the seventh postulate: the issue of consistency or reproducibility of results. Further reformulations were proposed by Lipkin [22] in 2013 in the Levels of Certainty in pathogen discovery and Byrd and Segre [23] in 2016 to advocate genetic sequencing of all members of a “microbial community” for classification and definitive identification.

## 6. A Proposed Framework for Assessing Transmission Causality of Respiratory Viruses 

Certainty (or diminution of uncertainty) is essential for all Gwaltney’s and Hendley’s postulates, especially on the crucial issues of correct identification of those who have an active infection and are contagious, i.e., capable of infecting contacts. This simple concept has been lost in a plethora of information overload of hurried low-quality studies in the current pandemic (in line with Huebner’s prescient observations). Clinical details of the exposed patient and a chain of transmission with laboratory confirmation of active infection in the exposed susceptible person are necessary to reduce the uncertainty over transmission. 

Across 591 studies, we identified laboratory confirmation of active infection as a critical result for interpreting the circumstances and finality of transmission. However, our reviews [9,10,11,12,13] show very heterogeneous methods of confirmation with low quality of evidence: from binary PCR to serial viral culture done with little context and even less rigour reported in the studies to assess the transmission. Few studies reported testing within a clinical context. For example, time from onset of symptoms and their severity are intimately tied to the likelihood of contagiousness (Table 6). 

Overall, we found that none of the 591 studies carried out serial viral cultures; of the 379 studies which used PCR as a diagnostic tool, 121 (32%) reported a cycle threshold and only 48 (13%) attempted to correlate its significance with viral culture (Table 6).

Few reported a positive culture and the culture methods used. This is an undesirable approach, but in the absence of evidence, such an argument at least requires the extensive use of positive controls.

Another review of studies of viral cultures for SARS-CoV-2 infectious potential assessment found that only 11/29 studies reported date of symptom onset in relation to the date of specimen collection [24].

To efficiently use all possible information and reduce uncertainty over the interpretation of the mode of transmission, we built a four-level hierarchical structure, similar to the levels of evidence used in evidence-based medicine [25], based on the work of the many scientists already mentioned who have contributed so much to causality and transmission.

**Level 1:** Binary PCR or nucleic acid antigen result in the absence of clinical data.

Lack of agent specificity means there are often no distinctive signs or symptoms readily attributable to a specific respiratory virus; reporting only PCR or other nucleic acid amplification tests as binary positive/negative does not identify current active infection. 

At the lowest level of confirmation, residual non-infectious RNA or DNA of the responsible viruses can be harboured and shed for weeks to months [24,26,27]. 

**Level 2:** Single Quantitative PCR result.

Adding further information, such as log concentrations of virus or PCR cycle threshold (Ct) values, reduces some of the uncertainty for identifying those with active infection but is not definitive. Current evidence shows that the lower the Ct representing a higher concentration of virus particles, the more likely a “positive” has infectious potential [24,26]. Contagiousness is also highest during the first few days from the onset of symptoms for many respiratory viruses, including SARS-CoV-2. In addition, immune-suppressing treatments or conditions may prolong infectiousness time [28], demonstrating the need for accurate clinical anamnestic data and a drug history to interpret the results of PCR.

**Level 3:** Cytopathic Effects (CPE).

More certainty is provided by cell culture and observation of the effects of inoculation of specimen washouts on the cells. Observed structural changes in host cells caused by viral invasion leading to visible cell lysis and other cytopathic phenomena are further indicative of viral replication. Clinically the patient may be still symptomatic, but the viral concentration may be too low to infect, and the CPE may be classed as weak. Therefore, coinfection or contamination cannot be fully ruled out. For example, Lednicky et al. [29] reported general virus-induced CPE within two days post-inoculation of specimens from a COVID-19 patient. However, PCR tests for SARS-CoV-2 RNA from the cell culture were negative; three other respiratory viruses were identified: influenza A H1N1, H3N2, and human coronavirus OC43. The exclusion of contaminants or co-infections in the plaques and serial culturing of similar specimens with the same results across different laboratories is required to reduce uncertainty further. 

**Level 4:** Serial Viral Culture or validated viral load surrogate and genome sequencing.

Immunohistochemistry directed at specific viruses and genome sequencing excludes co-infection and indicates a higher probability of the correct identification of the agent and viral similarity between donor and recipient, which are crucial for determining proof of transmission. Here again, accurate contact history and the clinical picture will provide chronological evidence of the likely chain of transmission. Patients within the early incubation period of the relevant respiratory virus, or of symptom onset, severe symptoms or pre-existing pathologies fall in this category. Sequencing does not require the presence of a viable virus, so only its coupling with repeated serial cultures will diminish the uncertainty around both transmission and causation. The presence of replicating virus, the absence of co-infection, and the similarity of the viral lineage of the presumed source gives rise to the strongest evidence of transmission and causation. 

During the SARS-CoV-2 pandemic an increasing body of evidence has shown the close relationship between viral load (e.g., copy numbers and cycle thresholds) and the likelihood of culturing replication competent virus. A decreasing Ct meaning an increasing genome copy number point to the high likelihood of the donor’s infectiousness without recourse to viral cultures. Such a relationship is reinforced if PCR methods are validated and harmonised on a national of subnational basis to increase consistency [30,31].

To prove transmission from one person to another requires two viral sequences to share an otherwise statistically improbable number of identical mutations. Genome sequences are required to exclude the possibility that the second person was infected by another circulating virus from elsewhere in the environment. The quantity of genome needing sequencing depends upon the spontaneous mutation rate (the rate of drift of the sequences). Ideally, these would be “silent” mutations (random sequence noise) unconnected to phenotypes subject to selection (for example, by the administration of antivirals).

To document how the second person became infected requires evidence that they were exposed in their environment (the route) and that the source was sufficiently infectious (low Ct) with infectious material (cultivatable virus) capable of transmitting via one or more routes to another person. Availability of all such evidence provides a high standard of proof of transmission in the age of genomics.

Our hierarchical Framework for Assessing Transmission Causality of Respiratory Viruses is synthesized in the Figure 1. 

As the next higher level of evidence is reported, clinical and transmission relevance increase with more reliable detail added, as symbolised by the arrows on the right. As reliability increases so does credibility and reproducibility, forming a hierarchy symbolised by the pyramid in the background.

The hierarchy allows an understanding of transmission evidence across our reviews and its uncertainties. Across our five reviews, the results interpreted by using our proposed framework show a lack of higher-level evidence. 

The hierarchical framework could be used in any situation where viral transmission needs investigating and reporting, especially with new emerging viral infections. Given the fleeting nature of some respiratory infections and the narrow window in which to study transmission events, a generally accepted framework would require less time in devising study design and more could be devoted to investigating and reporting. If some parts of the transmission chain cannot be documented, authors could indicate as such and reach tentative conclusions and identify existing gaps.

## 7. Discussion

Assigning causation relies on a consistent body of evidence which either excludes other biologically plausible alternative explanations for the events or makes them unlikely. Experimentation and consistency alongside our proposed framework provide a chain of evidence that can reduce the uncertainty over the transmission of respiratory viruses and increase the level of confidence in specific modes of transmission. The addition of a sixth postulate to Gwaltney–Hendley’s work requires the validation of the transmission chain by independent replication of results. To date at least one SARS-CoV-2 challenge study has been carried out [32]. Its results in terms of inoculum used, attack rate and development of symptoms are strikingly similar to the initial human challenge experiments conducted over five decades ago in the UK Common Cold Unit using the alpha coronavirus strain 229E [33].

Evidence that follows Bradford Hill’s principles for assessing causation: strength, temporality and experimentation (Gwaltney’s fifth postulate) along with independent replication and the hierarchy of transmission outcome events provides a coherent framework for assessing causation for the transmission of respiratory viruses. Continued uncertainty and lack of understanding of the major mode(s) of transmission and their respective frequency in any given setting with respect to SARS-CoV-2, underscores the critical importance of applying a framework for assessing causality in transmission of respiratory viruses. Given the current global COVID-19 pandemic and that more than two years since its historic arrival as a “modern-day plague”, we still do not have the fundamental questions definitively answered regarding the mode(s) and frequency of transmission, whether multiple modes can co-exist in the same setting, nor the minimal infective dose or the serological correlates of infection. It is imperative to provide high-quality evidence so that optimal intervention measures may be applied to achieve prevention of lives lost and protection against global economic and social demise. 

## Figures and Tables

**Figure 1 viruses-14-01605-f001:**
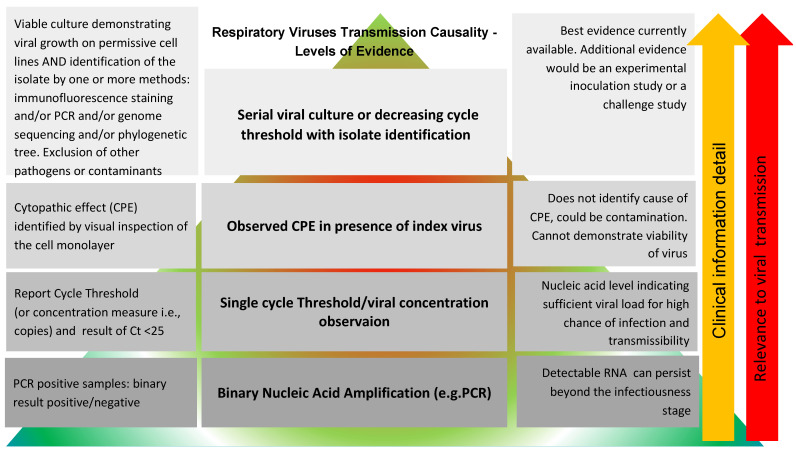
Levels of evidence for proof of the microbiological and clinical aspects of transmission of a viral respiratory pathogen. Decreasing cycle threshold over time refers to a validated surrogate marker. Key: CPE = cytopathic effect; PCR = polymerase chain reaction.

**Table 1 viruses-14-01605-t001:** Koch and Löffler 1884 postulates in their original formulation [14].

The microorganism must be found in abundance in all organisms suffering from the disease but should not be found in healthy organisms;
2. The microorganism must be isolated from a diseased organism and grown in pure culture;
3. The cultured microorganism should cause disease when introduced into a healthy organism;
4. The microorganism must be reisolated from the inoculated, diseased experimental host and identified as being identical to the original specific causative agent.

**Table 2 viruses-14-01605-t002:** Robert Huebner’s 1957 Bill of Rights for Prevalent Viruses [1].

(1) Isolation of a virus in culture;
(2) Repeated recovery of the virus from human specimens;
(3) Antibody response to the virus;
(4) Characterization and comparison with known pathogenic viruses;
(5) Constant association of the virus with specific illness;
(6) Reproduction of clinical illness in volunteer challenge studies;
(7) Epidemiologic studies (with controlled longitudinal studies offering the greatest value);
(8) Prevention of disease by vaccination.

**Table 3 viruses-14-01605-t003:** Gwaltney’s and Hendley’s proposed postulates for respiratory virus transmission [19].

**Postulate Number One** Microbial growth at the proposed anatomic site of origin.
**Postulate Number Two** Microbes present in secretions or tissues shed from the site of origin.
**Postulate Number Three** Microbes contaminate and survive in or on environmental substance or object.
**Postulate Number Four** Contaminated substance or object reaches portal of entry of new host.
**Postulate Number Five** Interruption of transmission by hypothesized route reduces incidence of natural infection.

**Table 4 viruses-14-01605-t004:** Evans’ 1991 list of “Five Realities of Acute Respiratory Disease” [17].

(1) The same syndrome could be produced by several agents;
(2) The same virus could produce several clinical syndromes;
(3) The cause of the syndrome varied by geographic area, age, and other factors;
(4) The causes of only about half of the common acute respiratory and intestinal syndromes and of about one-quarter of acute viral infections of the central nervous system have been identified;
(5) Diagnosis of the etiological agent could rarely be made on clinical grounds alone and required laboratory methods such as isolation of the virus and/or demonstration of an antibody response.

**Table 5 viruses-14-01605-t005:** Fredricks and Relman’s 1996 proposed reformulation of the original Koch’s postulates [21].

(1) A nucleic acid sequence belonging to a putative pathogen should be present in most cases of an infectious disease. Microbial nucleic acids should be found preferentially in those organs or gross anatomic sites known to be diseased (i.e., with anatomic, histologic, chemical, or clinical evidence of pathology) and not in those organs that lack pathology;
(2) Fewer, or no, copy numbers of pathogen-associated nucleic acid sequences should occur in hosts or tissues without disease;
(3) With resolution of disease (for example, with clinically effective treatment), the copy number of pathogen-associated nucleic acid sequences should decrease or become undetectable. With clinical relapse, the opposite should occur;
(4) When sequence detection predates disease, or sequence copy number correlates with severity of disease or pathology, the sequence–disease association is more likely to be a causal relationship;
(5) The nature of the microorganism inferred from the available sequence should be consistent with the known biological characteristics of that group of organisms. When phenotypes (e.g., pathology, microbial morphology, and clinical features) are predicted by sequence-based phylogenetic relationships, the meaningfulness of the sequence is enhanced;
(6) Tissue-sequence correlates should be sought at the cellular level: efforts should be made to demonstrate specific in situ hybridization of microbial sequence to areas of tissue pathology and to visible microorganisms or to areas where microorganisms are presumed to be located;
(7) These sequence-based forms of evidence for microbial causation should be reproducible.

**Table 6 viruses-14-01605-t006:** Virological and genomic evidence reported in 591 studies included in five systematic reviews of transmission of SARS-CoV-2. Key: Ct = cycle threshold; CPE = cytopathic effect.

Review	Primary Studies	PCR Result (% of Studies)	Ct (% of Studies)	Ct < 25 (% of Studies)	Attempted Viral Culture (% of Studies)	CPE (% of Studies)	Genome Sequencing (% of Studies)	Serial Viral Culture Positive (% of Studies)
Airborne Transmission [10]	127	53 (79.1%)	51 (40.2%)	5 (3.9%)	26 (20.4%)	5 (3.7%) ^1^	6 (4.7%)	3 (2.3%) ^2^
Fomite Transmission [11]	63	51 (81.0%)	13 (20.6%)	3 (4.8%)	11 (17.5%)	0	0	0
Orofecal Transmission [9]	77	46 (59.7%)	22 (28.6%)	7 (9.1%)	6 (7.8%)	1 (1.3%) ^3^	1 (1.3%)	0 ^3^
Close Contact Transmission [12]	258	163 (73.7%)	26 (10.1%)	6 (2.3%)	4 (1.6%)	2 (0.6%)	18 (5.8%)	2 (1.2%)
Vertical Transmission [13]	66	66 (100%)	9 (13.6%)	2 (3.0%)	0	0	1 (1.5%)	0
(% of primary studies)	591	379 (64.1%)	121 (20.5%)	23 (3.9%)	48 (8.1%)	9 (1.5%)	26 (4.4%)	5 (0.85%)

^1^ Some studies observed presumed virus-induced CPE. ^2^ Two studies detected other viruses, all studies had methodological limitations. ^3^ CPE did not show plaques and is not immunostained.

## Data Availability

All data included in the review are provided in the tables and text.

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
