# Peer review of "A Hierarchical Framework for Assessing Transmission Causality of Respiratory Viruses"

_viruses, 2022, doi:10.3390/v14081605_

Round 1
Reviewer 1 Report
Thank you for the opportunity to peer review your work. Providing a framework for assessing transmission causality of respiratory viruses is an essential component for further studies to accurately define modes of transmission and transmission dynamics of SARS-CoV-2. Much of the current controversies are related to this lack of an underlying definition or framework. I appreciated the historical context provided and how continuous updating has taken place to account for modern virology and molecular techniques. Your proposed framework certainly makes sense to me though I believe myself to have a relatively good grasp of respiratory virology and infectious diseases. The thought process behind the framework's generation and review of the existing literature to determine limitations is appropriate. I have no specific suggestions and enjoyed reading your work. Thank you again.
Author Response
Dear reviewer, your kind comments in this era of uncertainty and polarisation were like a breath of fresh air. We are trying our best to suggest general rules for the framework and are happy to accept any suggestions to improve the work in the future. We also hope that our framework will be followed and further developed and refined with use in the future. We have spelled check as suggested and thank you again for your support.
Reviewer 2 Report
This study by Jefferson et al builds on the results obtained from 4 systematic reviews of 591 studies on the different modes of transmission for SARS-CoV-2. The authors discuss the different methods used in assessing virus transmission for SARS-CoV-2 across 591 studies. The authors conclude that there was little evidence for causation mainly due to inconsistency and poor study design. Thus, the aspects of virus transmission are still unclear. Subsequently, the authors offer a new framework that should be used consistently for studying virus transmission. The framework consists of 4 levels, binary PCR output, viral concentration output, CPE observation, and viral culturing. The authors believe this framework allows for a more reproducible chain of evidence for transmission of respiratory viruses.
The study is overall well-presented and well written, although, it could improve by adding more details and clarifications at part. My main issue is I didn’t seem to fully understand the intent of the framework, i.e. when it should be applied. Do the authors suggest that any study investigating transmission chain events should follow the framework or only for new virus pathogens? I agree with the authors on the need for a more consistent method for investigating transmission, but I suspect that the reason most study use different methods is simply the lack of data and resources.
PCR and viral culture are labour intensive and not always accessible.
Respiratory infections are usually short and thus, sampling at a ‘transmission event’ is highly unlikely, limiting the collection of ‘evidence’.
Minor comments
Line 38. Is there really a ‘lack of high-quality evidence’? or just few studies?
Line 73. Ref 16 (Shutler et al) seems the wrong reference here.
Lines 169ff. I did not understand what the authors meant by ‘reached the level of viral culture’.
Figure 1. I was a bit confused by the layout. What is the meaning of the triangle in the background? And the two arrows in red and yellow?
Lines 251. I feel this needs more detail
